# Using mobile phone data to reveal risk flow networks underlying the HIV epidemic in Namibia

Eugenio Valdano [1], Justin T. Okano[1], Vittoria Colizza[2], Honore K. Mitonga [3] & Sally Blower [1✉]

Twenty-six million people are living with HIV in sub-Saharan Africa; epidemics are widely dispersed, due to high levels of mobility. However, global elimination strategies do not consider mobility. We use Call Detail Records from 9 billion calls/texts to model mobility in Namibia; we quantify the epidemic-level impact by using a mathematical framework based on spatial networks. We find complex networks of risk flows dispersed risk countrywide: increasing the risk of acquiring HIV in some areas, decreasing it in others. Overall, 40% of risk was mobility-driven. Networks contained multiple risk hubs. All constituencies (administrative units) imported and exported risk, to varying degrees. A few exported very high levels of risk: their residents infected many residents of other constituencies. Notably, prevalence in the constituency exporting the most risk was below average. Large-scale networks of mobility-driven risk flows underlie generalized HIV epidemics in sub-Saharan Africa. In order to eliminate HIV, it is likely to become increasingly important to implement innovative control strategies that focus on disrupting risk flows.

[1] Center for Biomedical Modeling, The Semel Institute for Neuroscience and Human Behavior, David Geffen School of Medicine, University of California, Los Angeles, CA, USA. [2] INSERM, Sorbonne Université, Institut Pierre Louis d'Epidémiologie et de Santé Publique, IPLESP, Paris, France. [3] Department of Epidemiology and Biostatistics, School of Public Health, University of Namibia, Windhoek, Namibia. ✉email: sblower@mednet.ucla.edu

Twenty-six million people live with HIV-infection in sub-Saharan Africa (SSA)[1]. In this region, HIV epidemics are generalized and populations are highly mobile[2–4]. However, global HIV elimination strategies do not take mobility into consideration. Multiple phylogenetic and epidemiological studies[5–12] of HIV in SSA have shown the importance of mobility for the dispersal of HIV, and the effect of mobility on HIV transmission. Individuals travel and have sex partners who live outside their home communities. Therefore, high levels of mobility can change who is at risk of infection, who transmits infection and where transmission occurs. Due to the difficulty of collecting data on mobility at a large spatial scale, previous studies[5–12] have focused on small geographic areas. Here, we study the impact of mobility on a high-prevalence HIV epidemic in SSA at the national level. We use a large-scale mobility network (constructed using Call Detail Records (CDRs) from mobile phones) that spans an entire country: Namibia (Fig. 1a). Worldwide, Namibia has one of the worst HIV epidemics; in 2017, the most recent survey found HIV prevalence to be ~15% in women and ~8% in men 15–49 years old[13]. HIV treatment has been rolled out in all countries in sub-Saharan Africa, and coverage levels are now high[13–15]. The effect of HIV treatment on transmission is now well known[16–18]. Here, our objective is to obtain an understanding of the effect of mobility on HIV transmission. Therefore, we conduct a retrospective analysis, and evaluate the effect of mobility on Namibia's HIV epidemic a decade ago, before HIV treatment was widely available. Our results lead to a new conceptual understanding of the dynamics of generalized HIV epidemics in SSA. We discuss the importance of our results for guiding current HIV elimination strategies.

Namibia has a population of ~2.5 million people, and the second lowest population density in the world. The country is divided into three administrative levels: 14 geographic regions, 54 provinces, and 121 constituencies. Due to its economy, Namibia has a highly mobile population. The economy is heavily dependent on mining, fisheries, large-scale farming and high-end tourism[19]. This has led to a system of circular migration of labor to mines, ports, farms, urban areas, and tourism nodes. Both women and men move around the country seeking employment; the most recent data show that ~70% of Namibians, 15 years or older, are economically active[19].

CDRs have an unmatched potential to measure population movements, and have been shown to be a key tool for retrospective analysis[20]. Over the past decade, CDRs have increasingly been used in health research[20–31]; particularly to model the geographic spread of infectious diseases[32], most recently SARS-CoV-2[22,28]. In these studies, population mobility networks are modeled by using algorithms to aggregate large datasets of CDRs into Origin-Destination (OD) matrices. The CDRs that we use have previously been used by Ruktanonchai and colleagues to evaluate the impact of mobility on malaria transmission in Namibia[29], and to design novel malaria elimination strategies[30,33]. We use the OD matrix that they constructed[34]; it contains aggregated CDRs from 9 billion calls/texts made from 1.19 million unique SIM cards over a 12-month period (October 2010 to September 2011) in Namibia[29]. The CDRs had been aggregated using an algorithm that accounts for the time spent by each individual in each constituency. In the OD matrix, rows specify the constituencies where individuals live, columns specify the constituencies they visit, and the entries/coefficients in the matrix specify weights. The weights are estimates of the average amount of time residents of one constituency spent in another constituency over 1 year. It is not possible to construct gender-specific OD matrices, because CDRs are collected anonymously. Therefore, we used the same OD matrix for both genders. To check whether this was an appropriate assumption we conducted two analyses. First, we determined whether one gender was more likely than the other to have owned or used a mobile phone. Second, we determined if there were gender differences in travel behavior.

A mobility network based on CDRs only reflects the mobility of individuals who use mobile phones. By 2012, 93% of Namibians (18 and older) were using mobile phones; therefore, the mobility network that we use reflects the mobility of a very high percentage of the population. Namibians who did not use mobile phones, in comparison with those who did, were more likely to live in very rural areas, have a low level of education, be unemployed, and older than 55. The movements of these individuals (who constitute 7% of the population) are not included in the mobility network. Conversely, the movements of some non-residents (e.g., tourists or business travelers)—if they had bought a local SIM card—are included in the mobility network. Given that Namibia had a population of 1.34 million individuals (15 or older)[35] and a very high percentage of the population were using mobile phones (with some owning more than one SIM card[36]), relatively few of the 1.19 million SIM cards in the database would have been owned by non-Namibians. Consequently, the movements of non-residents would have had little effect on the mobility network.

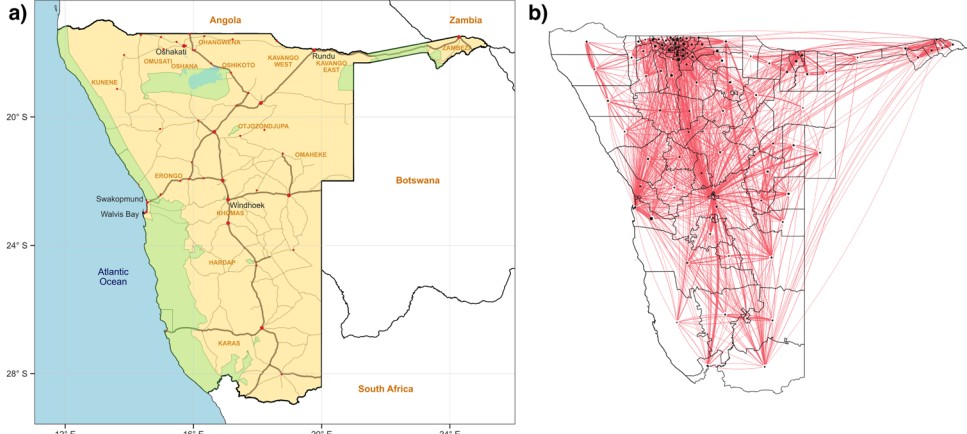

**Fig. 1 Geography of Namibia and mobility network. a** Map of Namibia. Red dots indicate main cities and towns. The names of the five largest cities appear in black. Region names appear capitalized in orange. Names of bordering countries appear in orange in bold. National park areas are indicated in green, water bodies in light blue. National borders appear in black. Main road (thick) and secondary roads (thin) are indicated in brown. **b** Map of mobility network, constructed from CDRs, based on an OD matrix constructed by Ruktanonchai and colleagues[34]. Red lines show travel between constituencies.

We use an approach based on modeling spatial networks to determine the effect that mobility had on the risk of acquiring, transmitting, and dispersing HIV in the time before treatment was made widely available. We conceptualize HIV epidemics as spatial networks consisting of communities that import and/or export risk; communities are interconnected by groups of individuals moving, on a temporary basis, between them. Mobility is specified by the OD matrix. A community imports risk if their residents are at-risk of being infected with HIV by residents of other communities. Risk can be imported into a community by either of two mechanisms: (i) their uninfected residents visit other communities and acquire HIV, or (ii) HIV-infected residents of other communities visit and transmit HIV. The greater the flow of imported risk to a community, the more vulnerable the community is to the risks posed by the other communities. The most vulnerable communities are called in-flow risk hubs. A community exports risk if their residents have the potential to infect residents of other communities. Risk can be exported from a community by either of two mechanisms: (i) their HIV-infected residents visit other communities and transmit HIV, or (ii) uninfected residents of other communities visit and acquire HIV. The greater the flow of exported risk from a community, the more "risky" the community is to the other communities. The most risky communities are called out-flow risk hubs.

In our spatial network modeling framework, each node represents a constituency, and each link represents a mobility-driven risk flow between two constituencies. Risk flows are gender-specific: the risk of acquiring HIV for women depends upon the prevalence of HIV in men, and the risk of acquiring HIV for men depends upon the prevalence of HIV in women. Our framework enables us to quantify and visualize countrywide networks of mobility-driven risk flows, determine the drivers of risk flows, detect in-flow and out-flow risk hubs, and find the geographic location of every risk hub in Namibia. The residents of constituencies that are in-flow risk hubs are at the greatest risk of acquiring HIV from individuals, who live outside their constituency. The residents of constituencies that are out-flow risk hubs have the highest probability of transmitting HIV to individuals, who live outside their constituency. A mathematical description of the spatial risk flow networks is given in the methods.

## Results

**Gender and mobility**. We did not find gender differences among adults in Namibia in the ownership of mobile phones (84.3% (women), 84.2% (men)), the usage of mobile phones (93.0% (women), 92.8% (men)), or in the frequency of phone calls and texting (Supplementary Table 1). We did find that women (15–49 years old) were slightly less likely to have taken overnight trips than men: 37% of women versus 43% of men (Supplementary Table 2). However, similar proportions (18%) of each gender took trips that lasted for more than a month.

**The effect of mobility on changing risk**. The mobility network (Fig. 1b, which is a visualization of the OD matrix), shows that the population of Namibia was highly mobile, and both short and long distance trips were common. On average, residents spent 2.6 months per year outside their home constituency between October 2010 and September 2011, although there was considerable geographic variation, ranging from approximately 1–6 months (Fig. 2a and Supplementary Fig. 1). As a result of the specific travel patterns, many of the constituencies were highly connected (Fig. 1b). In 2010, there were 107 constituencies in Namibia; 11 of these are not represented in the mobility network

because, due to a low population density, they lacked cell phone towers.

To conduct our analyses, we needed to estimate the prevalence of HIV in women and men at the approximate time that the CDRs were collected. However, the first estimate of HIV prevalence in Namibia, based on a representative sample of the population, was based on data collected in 2013[37]. At that stage of the epidemic, HIV prevalence was relatively stable[38]. Therefore, we made the parsimonious assumption that prevalence did not change substantially between 2010 and 2013. In 2013, the average prevalence in women and men (aged 15–49 years old) was 17 and 11%, respectively[37]. However, there was considerable geographic variation: ranging from 6 to 39% in women (Fig. 2b), and from 0 to 24% in men (Fig. 2c). Notably, the prevalence patterns were considerably different for men and women.

The connectivity-prevalence matrix (Fig. 2d) shows the effect that the mobility network (Fig. 1b) had on changing the risk of acquiring HIV for men. Data below the diagonal show the proportion of men who spent time in constituencies, where the prevalence of HIV in women was higher than in their home constituency; these men potentially increased their risk of acquiring HIV. Data above the diagonal show the proportion of men who spent time in constituencies, where the prevalence in women was lower than in their home constituency; these men potentially decreased their risk of acquiring HIV. Data along the diagonal show the proportion of men who spent the majority of their time in their home constituency, and in other constituencies, where prevalence in women was similar to that in their home constituency. The level of risk for these men essentially did not change. We found similar results for women (Supplementary Fig. 2). Taken together, our results imply that there were substantial mobility-driven risk flows, for both genders, among different areas of Namibia in the time before treatment was widely available.

**Risk flow networks: risk importation**. We found that all constituencies imported risk to some degree. Imported risk for men, and the mechanism by which it was imported, is shown in Fig. 3a: orange data show the risk that was imported into each constituency due to travel by its uninfected male residents, green data show the risk that was imported into each constituency due to visits from HIV-infected women, who lived in other constituencies. Stacked values (orange plus green data) show the total amount of risk that was imported into each constituency. Constituencies in Fig. 3a are ordered based on HIV prevalence in women: from the lowest (6%) to the highest (39%). Supplementary Fig. 3 shows similar results, as in Fig. 3a, for women.

We defined in-flow risk hubs as constituencies that were in the top 40% of constituencies in terms of imported risk: 38 constituencies were risk hubs. Each risk hub has a different amount of risk flowing into it; the greater the in-flow of risk, the more important the constituency is as a risk hub. Some constituencies (e.g., Oshakati East: blue dot, Fig. 3a) were in-flow risk hubs for men, primarily because they were visited by HIV-infected women who lived in other constituencies (green data). Other constituencies (e.g., Kongola: red dot, Fig. 3a) were in-flow risk hubs for men, primarily because of travel by their uninfected male residents (orange data).

The in-flow risk network for men in Oshakati East (Supplementary Fig. 4) shows which constituencies the female visitors were from, and the magnitude of the in-flow of risk from each constituency. Risk was imported because many of the female visitors were from constituencies, where HIV prevalence in women was higher than in Oshakati East.

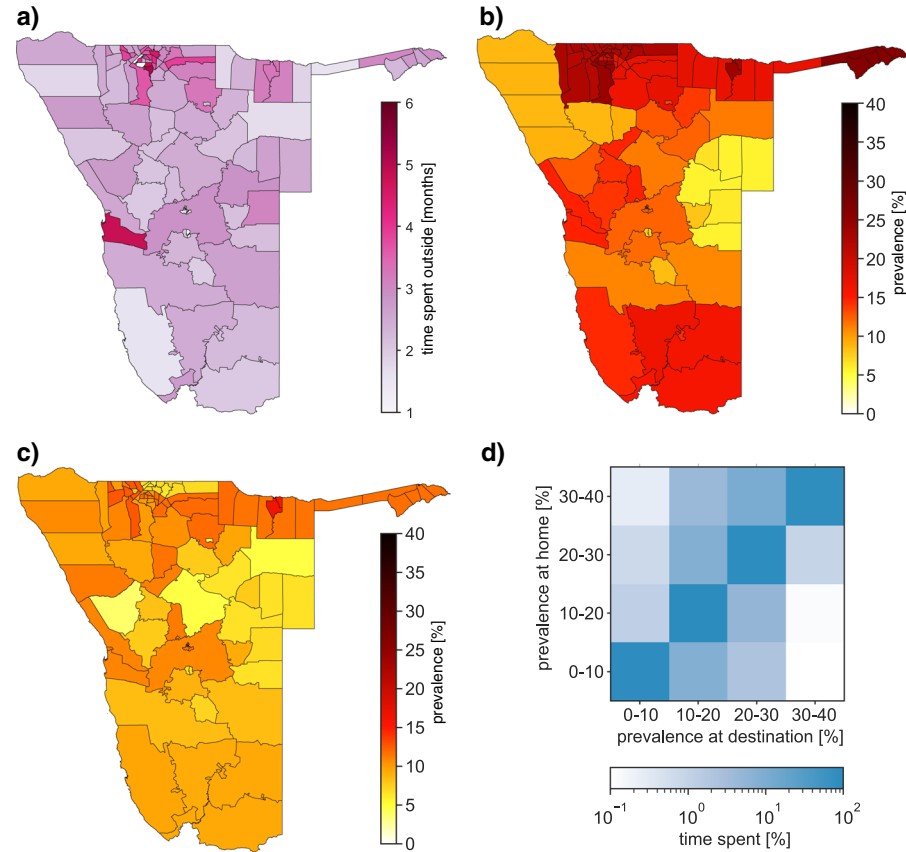

**Fig. 2 Spatial relationships for mobility and HIV prevalence. a** Cartographic map showing the average proportion of time (over a year: 2010–2011) residents spent outside their home constituencies. **b** HIV prevalence map showing geographic variation in prevalence in women (15–49 years old) at the level of the constituency in 2013. **c** HIV prevalence map showing geographic variation in prevalence in men (15–49 years old) at the level of the constituency in 2013. **d** Matrix showing the proportion of time (over a year) that men in Namibia spent in each prevalence "class", as a function of the prevalence class of their home constituency. Prevalence refers to prevalence in women. The matrix is color-coded to show the fraction of time spent in the destination constituency. A logarithmic scale is used, ranging from 0.1 to 100%.

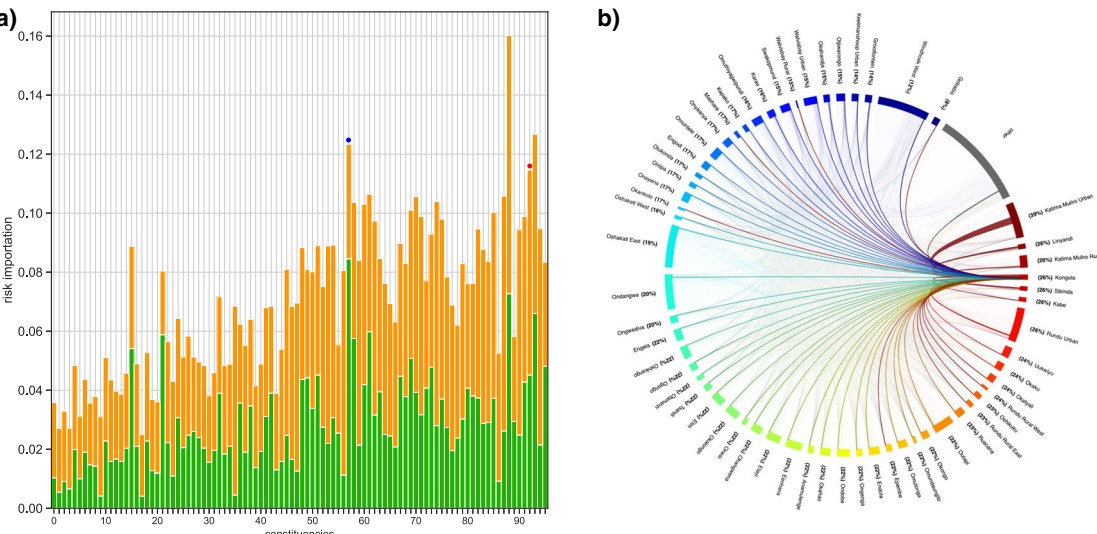

**Fig. 3 Importation of risk. a** Histogram showing the importation of risk for men into each constituency. The *y*-axis shows the value for imported risk; a mathematical definition of imported risk is given in the methods (Eq. 5). The numbers on the *x*-axis refer to specific constituencies; the key code is given in Supplementary Table 4. Constituencies are ordered by increasing HIV prevalence in women: from 6 to 39%. The constituency of Oshakati East is labeled with a blue dot, the constituency of Kongola with a red dot. Orange data show the risk for men that was imported into each constituency due to travel by their uninfected male residents. Green data show the risk for men that was imported into each constituency due to visits from HIV-infected women, who lived in other constituencies. The stacked value represents the total amount of imported risk. **b** Chord diagram showing the in-flow risk network for men in Kongola; the lines represent risk importation due to uninfected male residents of Kongola visiting other constituencies. The thickness of the lines is proportional to the amount of imported risk.

The in-flowrisk network for men in Kongola (Fig. 3b and Supplementary Fig. 5) shows which constituencies its male residents visited, HIV prevalence in women in those constituencies, and the magnitude of the in-flow of risk from each constituency. Even though prevalence in women in Kongola was extremely high (26%), risk was imported because male residents of Kongola visited a constituency (Katima Muliro Urban), where prevalence in women was even higher, 39% (Fig. 3b).

**Risk flow networks: risk exportation**. We found that all constituencies, as well as importing risk, exported risk to some degree. Exported risk for men, and the mechanism by which it was exported, is shown in Fig. 4a: orange data show the risk (for men) that was exported from each constituency due to visits from uninfected men who lived in other constituencies, green data show the risk (for men) that was exported from each constituency

due to travel by their HIV-infected female residents. Stacked values (orange plus green data) show the total amount of risk (for men) that was exported from each constituency. Constituencies in Fig. 4a are in the same order as in Fig. 3a. Supplementary Fig. 6 shows similar results, as in Fig. 4a, for women.

We defined out-flow risk hubs as the top 40% of constituencies in terms of exported risk: 38 constituencies were risk hubs. Each risk hub has a different amount of risk flowing out of it; the greater the out-flow of risk, the more important the constituency is as a risk hub. Notably, only a few out-flow risk hubs exported a great deal of risk. Windhoek West (blue dot, Fig. 4a) was the most important exporter of risk for men, even though prevalence in female residents of Windhoek West was well below the national average (12% vs. 17%). Windhoek West was the most important out-flow risk hub, because it was a very important mobility hub and many of the male visitors to Windhoek West were from constituencies, where prevalence in women was lower than 12%.

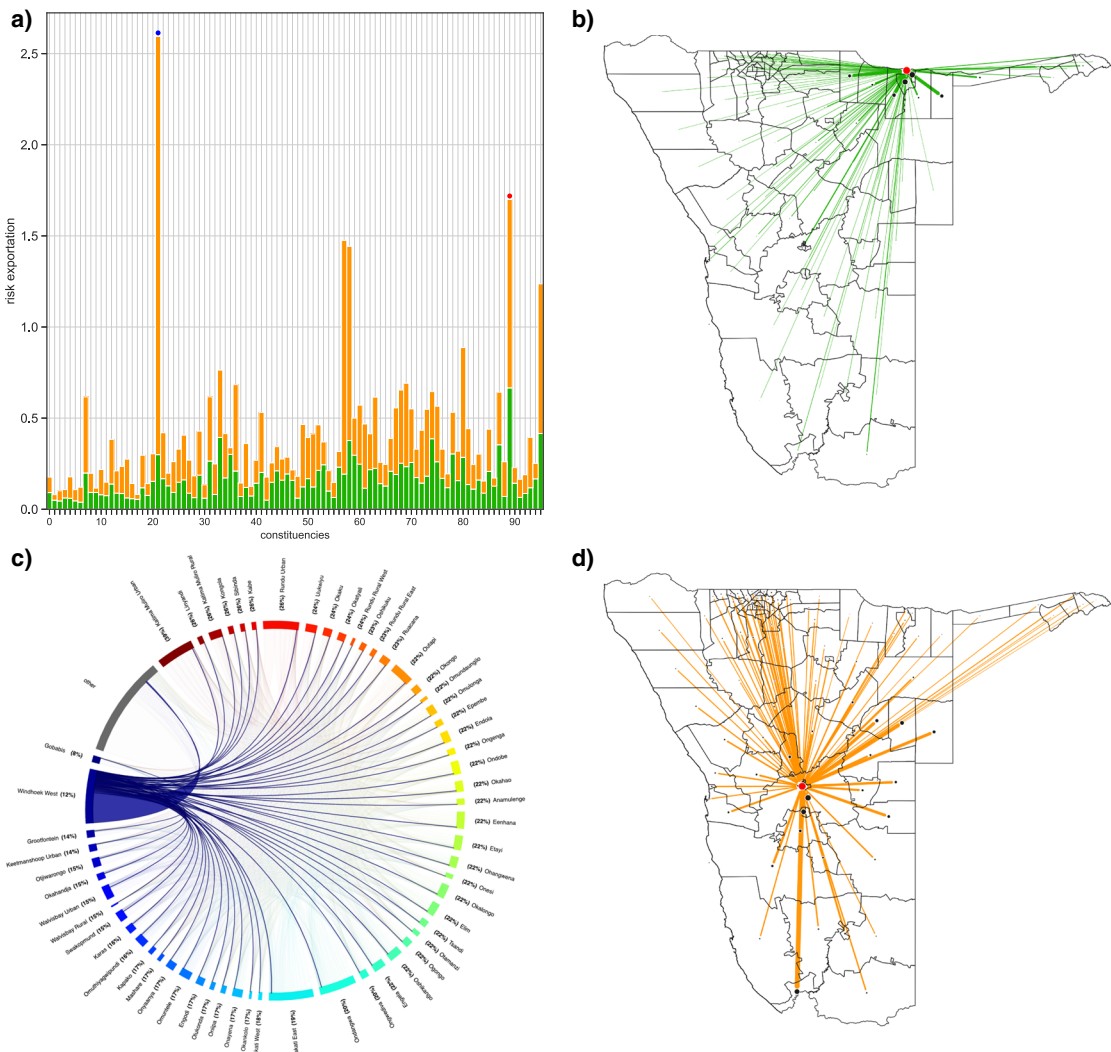

**Fig. 4 Exportation of risk. a** Histogram showing the exportation of risk for men from each constituency. The *y*-axis shows the value for exported risk; a mathematical definition of exported risk is given in the methods (Eq. 5). The numbers on the *x*-axis refer to specific constituencies; the key code is given in Supplementary Table 4. Constituencies are ordered as in Fig. 3a. The constituency of Windhoek West is labeled with a blue dot, the constituency of Rundu Urban with a red dot. Orange data show the risk for men that is exported from each constituency due to visits from uninfected men who lived in other constituencies. Green data show the risk for men that is exported from each constituency due to travel by their HIV-infected female residents. The stacked value represents the total amount of exported risk. **b** Map showing the out-flow risk network for men; risk is due to HIV-infected female residents of Rundu Urban (represented by the red dot) visiting other constituencies. **c** Chord diagram showing the out-flow risk network for men; risk is due to uninfected male residents of other constituencies visiting Windhoek West. **d** Map showing the out-flow risk network from Windhoek West (red dot). The thickness of the orange lines is proportional to the amount of exported risk.

Rundu Urban (red dot, Fig. 4a) is an example of an out-flow risk hub for men where risk was exported due to the travel behavior of their HIV-infected female residents (green data). Rundu Urban (Fig. 4b) exported risk to many constituencies, near and far; risk was exported because prevalence in women in Rundu Urban was much higher (26%) than in the constituencies the women visited. Other constituencies (e.g., Windhoek West) were out-flow risk hubs for men, primarily because they were visited by uninfected men who lived in other constituencies (orange data). The out-flow risk network from Windhoek West (Fig. 4c, d) shows which constituencies the male visitors were from, HIV prevalence in women in those constituencies, and the magnitude of the out-flow of risk to each constituency.

**Mobility-driven risk of acquiring HIV**. We used our risk flow networks to calculate the average overall annual risk of acquiring HIV in Namibia, subdivided into three components: risk due to localized transmission in an individuals' home constituency, risk due to visiting other communities, and risk due to residents of other communities visiting an individuals' home constituency. We define localized transmission as transmission between residents of the same constituency, when they are in their home constituency. We found that the majority of the risk (60%) was due to localized transmission, 25% of the risk was due to visiting other communities, and 15% of the risk was due to residents of other communities visiting an individuals' home constituency. Overall, ~40% of the total risk was due to mobility-driven transmission. We obtained similar results for men and women.

**Identifying the geographic location of risk hubs**. Figure 5a shows the geographic location of all out-flow risk hubs. The red data show the top 10% of constituencies in terms of exported risk, i.e., the most important out-flow risk hubs. Five of these risk hubs are constituencies that are very close together in the central part of the north, the other five risk hubs are widely dispersed throughout the country.

Figure 5b shows the geographic location of all in-flow risk hubs in Namibia. The red data show the top 10% of constituencies in terms of imported risk, i.e., the most important in-flow risk hubs. Several of the important in-flow risk hubs are close to important out-flow hubs in the central part of the country in the north, and in the Zambezi region in the northeast. Notably, two constituencies are very important as both in-flow and out-flow risk hubs: Oshakati East and Ondangwa, neighboring constituencies in the north (the only constituencies shown in red in both Figs. 5a and 5b).

## Discussion

The effect of treatment on generalized HIV epidemics in SSA is well known[16–18]. Previous studies have made important contributions to our understanding of the impact of mobility on transmission at the community level[5–12]. However, very little is known about the effect of mobility—at the population level—on these generalized epidemics. Here by reconstructing the state of the HIV epidemic in Namibia a decade ago (before treatment was widely available) and using CDRs to model population-level mobility, we were able to evaluate the effect of mobility on an HIV epidemic at the national-level. Using a modeling framework, we discovered that a combination of high levels of mobility, substantial geographic variation in prevalence, and specific travel patterns, had generated complex risk flow networks that spanned Namibia. The networks dispersed risk throughout the country: increasing risk in some areas, decreasing risk in others. All constituencies were vulnerable to the in-flow of risk, some more than others. Some risk flows were between constituencies that were close together, others were between constituencies that were geographically very far apart. All constituencies posed some degree of risk to other constituencies. A few constituencies were extremely risky; the most risky was Windhoek West, where prevalence was slightly less than the national average. Our results show that mobility has been extremely important for the acquisition, transmission, and dispersal of HIV in Namibia: overall, ~40% of the total risk of acquiring HIV was driven by mobility. More importantly, taken together, our results provide a new conceptual understanding of generalized HIV epidemics in SSA: i.e., they should be understood as large-scale complex networks of mobility-driven risk flows. Our results imply that, to eliminate HIV, it will be essential to design new control strategies that focus on disrupting these risk flows.

Many other countries in SSA have similar characteristics to Namibia: substantial geographic variation in the prevalence of HIV, a highly mobile population, and circular migration. Therefore, spatial risk flow networks are likely to exist in these countries, and throughout SSA. Networks will be country-specific: they will depend upon the mobility level of the population, the geographic variation in HIV prevalence, and the degree to which travel patterns link high-prevalence and low-prevalence areas. The importance of mobility on HIV epidemics can be expected to vary by country[39,40].

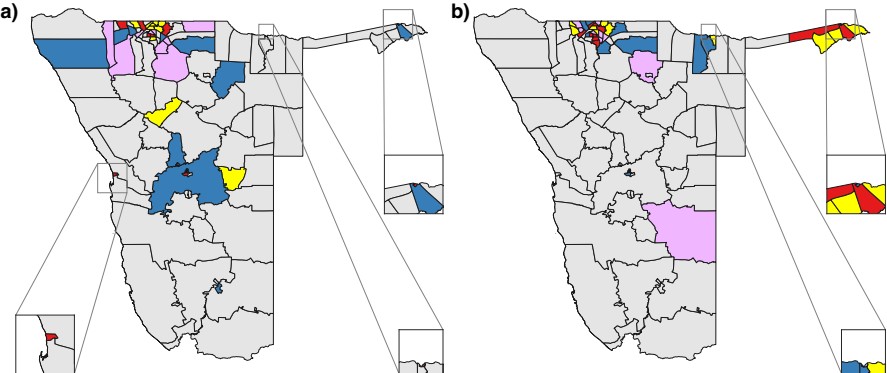

**Fig. 5 Geographic location of risk hubs. a** Map showing the risk hubs for exported risk, i.e., the out-flow risk hubs. The top 10% of constituencies, in terms of exported risk, are shown in red. The next 10% are in yellow, the next 10% in light purple, and the next 10% in blue. **b** Map showing the risk hubs for imported risk, i.e., the in-flow risk hubs. The top 10% of constituencies, in terms of imported risk, are shown in red. The next 10% are in yellow, the next 10% in light purple, and the next 10% in blue.

HIV treatment both increases survival, and prevents onward transmission[16–18]. HIV treatment in Namibia, as in other countries in SSA, was rolled out based on geographically targeting high-prevalence areas[41]; high-prevalence areas are defined as Hot-Spots. Hot-Spots were disproportionally allocated treatment. This targeting strategy was based on two assumptions, that: (i) risk was localized, and (ii) the highest prevalence areas were the areas of greatest risk. However, we have found that these two assumptions did not hold in Namibia; a substantial proportion (40%) of the risk was not localized, and high-risk areas (e.g., Windhoek West) were not always high-prevalence areas. Therefore, the geographic targeting strategy that was employed did not target several of the areas in Namibia, where the risk of acquiring and transmitting HIV was the greatest.

Our results provide qualitative insights into the effect that treatment had on the Namibian epidemic. Treatment would have disrupted the risk flow networks. In the targeted high-prevalence areas that were important out-flow risk-hubs, the structure of the risk flow networks would have spatially dispersed the beneficial effect of treatment on reducing transmission. This would have decreased both localized transmission and transmission in many other constituencies in Namibia. In the targeted high-prevalence areas that were not important out-flow risk-hubs, only localized transmission would have decreased.

Our results have important policy implications for the design of HIV elimination strategies in Namibia. The country has made considerable progress in controlling its HIV epidemic. Progress has been the result of political leadership, an effective community-centered approach to interventions, and the strategic expansion of treatment services. There are many preventative modalities that are currently being used successfully in HIV interventions in Namibia: e.g., pre-exposure prophylaxis, medical circumcision, and condoms[41]. The Government began nation-wide implementation of the "Treat All" policy in April 2017. This policy recommends providing treatment to all HIV-infected individuals, and removing treatment eligibility requirements. By 2017, the HIV treatment program in Namibia was doing extremely well: ~83% of all HIV-infected adults in Namibia were on treatment[13], a level of coverage among the highest in Africa[42]. The Government's current goals are to reach UNAIDS' 2030 treatment targets for elimination and to achieve treatment equity[41] with respect to geography, age, and gender targets (especially for young women). UNAIDS' goals for 2030 are to have 95% of HIV-infected individuals diagnosed, 95% of the diagnosed on treatment, and 95% of treated patients with viral suppression[43]. Our new understanding of the generalized HIV epidemic in Namibia suggests that, in order to eliminate HIV, control strategies will need to take into account the riskiness, and vulnerability, of constituencies. To design effective strategies, it will be necessary to identify the current mobility network; therefore more recent data on CDRs in Namibia need to be collected. Then, by expanding the methodology that we have presented in this study to include treatment, the current risk flow networks and the risk hubs can be identified. We recommend developing strategies that preferentially target the most important in-flow and out-flow risk hubs in the country. Within in-flow risk hubs, it is most important to target uninfected residents; within out-flow risk hubs, it is most important to target HIV-infected residents.

There are limitations to our study, both with respect to the data that we have used and the assumptions that we have made. We have made the parsimonious assumption that HIV prevalence did not change substantially between 2010 (the year the CDRs were collected) and 2013 (the year the prevalence data were collected). Although this assumption cannot be verified, it is supported by the fact that UNAIDS prevalence estimates for Namibia did not change significantly over this three year time period[38]. CDRs are always anonymized and therefore OD matrices cannot be disaggregated on the basis of gender or any other demographic factors[44]. In our study, as in all previous studies that have used CDRs to model population-level mobility[20–31], we assumed that neither phone ownership, usage, or travel behavior differed substantially between genders. The data that we have presented have shown that phone ownership and usage and length of trips did not differ by gender. However, men traveled slightly more frequently than women, and we do not know whether there were gender differences in the origins and destinations of trips. Therefore, it is possible that women and men had different mobility networks. If true, this could potentially have changed which of the constituencies were risk hubs. However, it would not have changed our central conclusions: (i) that mobility has been extremely important for the acquisition, transmission, and dispersal of HIV in Namibia, and (ii) that our results present a new conceptual understanding of generalized HIV epidemics in SSA. The CDRs used to construct the OD matrix show that there was seasonal variation in mobility. However, it is not necessary to model seasonal changes in mobility as HIV transmission occurs throughout the year. Even a substantial seasonal variation in transmission would have a negligible impact (over a year) on prevalence; this is because prevalence is an order of magnitude higher than incidence. Finally, our study focuses on identifying spatial networks of risk flows in a generalized epidemic where the vast majority of transmission is through heterosexual sex. We have not modeled spatial networks of risk flows among men who have sex with men. These networks could be included in future studies.

Many countries in SSA, including Namibia, have very mobile borders[45,46], therefore it is essential to develop more complex spatial network models that can be used to evaluate the effect of intra-country and inter-country mobility on generalized HIV epidemics throughout SSA. Many in-flow and out-flow hubs in Namibia are close to the Angolan border (Fig. 5a). HIV prevalence along the border (in adults aged 15–49 years old) is substantially lower in Angola (5–6%)[47] than in Namibia (9–32%). This suggests that the out-flow of risk from Namibia would have been greater than the in-flow of risk from Angola. Consequently, Namibia would have had more of an impact on the HIV epidemic in Angola, than Angola would have had on the HIV epidemic in Namibia.

We propose that high levels of mobility may be reducing the effectiveness of current epidemic control strategies in SSA. UNAIDS' HIV elimination strategy is based, in large part, on the assumption that a high coverage of treatment will reduce incidence to an extremely low level. Four large-scale clinical trials were set up to test this premise: the ANRS 12249 TasP study in South Africa[48], the SEARCH study in Kenya and Uganda[49], the HPTN 071 (PopART) study in South Africa and Zambia[50], and the Ya Tsie study in Botswana[51]. The first three of these trials failed to show a reduction in incidence, and the Ya Tsie study showed a modest decrease of ~30%[51]. We suggest that the high level of mobility in these countries could have been an important factor that contributed to the failure of these trials. High incidence rates in the areas of study could have been maintained due to the importation of risk. This could have occurred due to uninfected residents of communities in the study area visiting communities outside the study area, or due to HIV-infected residents of other communities visiting communities in the study area. As HIV approaches elimination, mobility-driven transmission is likely to become increasingly important. This has already been seen in elimination campaigns for polio[52] and malaria[53]; high levels of mobility have led to the continuous introduction of new infections into areas, where localized transmission had already been stopped or reduced to extremely low levels. Mobility may also become a substantial barrier to the elimination of HIV in SSA.

## Methods

**Assessing gender bias in mobility**. To determine whether there was gender bias in mobile phone ownership or usage we analyzed data collected in 2012 in Namibia in the 5th round of the Afrobarometer surveys[36]. To determine if there were gender differences in travel behavior, we analyzed data (stratified by gender) from the 2013 Namibian Demographic and Health Survey (NDHS)[37] on the number of overnight trips that had been made in the previous 12 months (between 2012 and 2013), and on the number of those trips that had lasted more than a month.

The Afrobarometer surveys are conducted in 37 African countries[36]. The survey utilizes a multistage clustered stratified sampling design to generate a representative sample of the population, 18 years and older. It collects public opinion data on a country's economy, governance, and society. The survey also collects data on ownership and usage of mobile phones.

The 2013 NDHS utilized a two-stage cluster design to collect a representative sample of Namibia's population[37]; clusters were georeferenced. Demographic and behavioral data were collected from 14,499 individuals in 9849 households. Participants who were 15 or older were tested for HIV infection. Each individual's test results were linked to their demographic and behavioral data. The overall response rate was high: 92% for women, 85% for men. Participation in HIV-testing (in 15–49 year olds) was also high: 83% for women, 74% for men.

**HIV prevalence mapping**. We used data from the 7731 individuals (aged 15–49 years old) who were tested for HIV in the 2013 NDHS[37] to construct the gender-specific prevalence maps for Namibia: Fig. 2b (women) and Fig. 2c (men). Prevalence was estimated, and maps constructed, using R v. 3.6.3 (packages "survey" and "sp").

**Calculating prevalence-connectivity matrices**. We combined the OD matrix with the prevalence data to construct the gender-specific prevalence-connectivity matrices. These matrices are shown in Fig. 2d (men) and Supplementary Fig. 2 (women). Each matrix was calculated by using the data in the OD matrix to determine the time spent in each constituency. The prevalence data for women were used to construct the prevalence-connectivity matrix for men; the prevalence data for men were used to construct the prevalence-connectivity matrix for women.

**Modeling spatial networks of risk flows**. To model these networks, we used three sources of data: (i) the OD matrix, (ii) HIV prevalence data[37], and (iii) Census data[35]. We define risk as the probability that a new sex partner, who is a resident of constituency $i$, is infected with HIV. If the population is not mobile, the risk to one gender simply equals the prevalence of HIV in the opposite gender. If the population is mobile, risk is calculated as described below.

The _effective population size_ ($\hat{n}_i$) and the _effective HIV prevalence_ ($\hat{p}_i, \hat{q}_i$) are calculated. The effective population size ($\hat{n}_i$) is the expected number of individuals present in a constituency at a given time. The effective HIV prevalence ($\hat{p}_i, \hat{q}_i$) is the expected prevalence of HIV in men and women, respectively, in a constituency at a given time.

$$\hat{n}_i = \sum_j n_j A_{ji}, \tag{1}$$

$$\hat{p}_i = \frac{\sum_j p_j n_j A_{ji}}{\hat{n}_i}, \tag{2}$$

$$\hat{q}_i = \frac{\sum_j q_j n_j A_{ji}}{\hat{n}_i}, \tag{3}$$

where

$$l_i^{\text{rel}} = \frac{l_i}{r_i},$$

$n_i$ represents the size of the resident population of constituency $i$.

$A_{ij}$ represents the average proportion of time a resident of constituency $i$ spends in constituency $j$ (measured over a year); by construction $\sum A_{ij} = 1$.

$p_i$, $q_i$ represent the HIV prevalence in men and the HIV prevalence in women, respectively, among the resident population of constituency $i$.

The total risk for women, $r_i$, is then calculated; $r_i$ is the probability that when a woman, who is a resident of constituency $i$, meets a new male sex partner, he is infected with HIV. The total risk for women in constituency $i$ is defined as follows:

$$r_i = \sum_j A_{ij} \hat{p}_j \tag{4}$$

The total risk can be broken down into three distinct contributions, which make up the right-hand side of Eq. (5):

$$r_i = l_i + \sum_{j \neq i} t_{ji}^F + v_{ji}^F, \tag{5}$$

where

$l_i$ **local risk for women**. This is the probability that a woman, who is a resident of constituency $i$, meets a new sex partner, she is in her home constituency, and the partner is infected with HIV and a resident of the same constituency $i$.

$t_{ij}$ **risk for women due to their travel**. This is the probability that a woman, who is a resident of constituency $j$, meets a new sex partner while visiting constituency $i$, and the partner is infected with HIV and a resident of constituency $i$.

$v_{ij}$ **risk for women due to travel of HIV-infected men who are residents of other constituencies**. This is the probability that a woman, who is a resident of constituency $j$, meets a new sex partner in her home constituency, and the partner is infected with HIV and a resident of constituency $i$.

$$l_i = \frac{(A_{ii})^2 p_i n_i}{\hat{n}_i}, \tag{6}$$

$$t_{ij} = A_{ji} \hat{p}_i, \tag{7}$$

$$v_{ij} = \frac{p_i n_i A_{ij} A_{jj}}{\hat{n}_j}. \tag{8}$$

Note that inserting Eqs. (6, 7, 8) in Eq. (5) leads to Eq. (4). By swapping $p$ for $q$ the risks for men can be calculated.

The relative impact of a risk on any specific constituency can be defined in terms of the relative impact on the total risk:

$$l_i^{\text{rel}} = \frac{l_i}{r_i}, \tag{9}$$

$$t_{ij}^{\text{rel}} = \frac{t_{ij}}{r_j}, \tag{10}$$

$$v_{ij}^{rel} = \frac{v_{ij}}{r_j}, \tag{11}$$

For instance, $t_{ij}^{\text{rel}} = 0.3$ means that 30% of the risk in constituency $j$ comes from constituency $i$, in terms of risk type $t$.

The risk networks are specified in terms of the flows of risk ($t_{ij}$ and $v_{ij}$) between constituencies, where constituencies are the nodes in the network. A link from constituency $i$ to constituency $j$ always represents a flow of risk from constituency $i$ to constituency $j$. Constituency $i$ increases the risk in constituency $j$ by adding $t_{ij}$ and $v_{ij}$ to $r_j$. Clearly either $t_{ij}$ or $v_{ij}$ or both can be zero, if no flow of risk occurs.

The risk flow networks are weighted and directed networks. Weights are specified in terms of the amount of risk that a link carries. The direction of risk flows is not always the same as the direction of the mobility fluxes that generate them. It is the same direction for $v$, which flows from constituency $i$ to constituency $j$ through individuals going from constituency $i$ to constituency $j$ (see the presence of the term $A_{ij}$ in Eq. (8)). It is the opposite direction for $t$, which flows from constituency $i$ to constituency $j$ through individuals going from constituency $j$ to constituency $i$ (see the presence of the term $A_{ji}$ in Eq. (7)). In this latter case, risk in constituency $j$ increases as its residents travel to other constituencies and risk getting infected there.

Risk was calculated using Python 3 and standard libraries (numpy v1.19, pandas v1.2).

**Calculating risk importation and exportation**. The in-strength of a node in a weighted, directed network is the sum over the weights of its incoming links. We define risk importation as the in-strength of a node in the networks. Therefore, the risk importation into each node is:

$$t_i^{\text{IN}} = \sum_{j \neq i} t_{ji}, \tag{12}$$

$$v_i^{\text{IN}} = \sum_{j \neq i} v_{ji}. \tag{13}$$

The out-strength of a node in a weighted, directed network is the sum over the weights of its outgoing links. We define risk exportation as the out-strength of a node in the networks of relative impacts:

$$t_i^{\text{OUT}} = \sum_{j \neq i} t_{ij}^{\text{rel}} = \sum_{j \neq i} \frac{t_{ij}}{r_j}, \tag{14}$$

$$v_i^{\text{OUT}} = \sum_{j \neq i} v_{ij}^{\text{rel}} = \sum_{j \neq i} \frac{v_{ij}}{r_j}. \tag{15}$$

Notably, $t_i^{\text{OUT}}$, $v_i^{\text{OUT}}$ can be larger than one. For instance, if constituency $i$ generates 60% of type-$t$ risk of $j$, and 50% of type-$t$ risk of $k$, then its risk exportation, $t_i^{\text{OUT}}$ is equal to 1.1.

We calculated the risk flows for each node/constituency in the networks; the values are shown in Supplementary Table 3 for women and Supplementary Table 4 for men. These values were also used to construct the histograms shown in Fig. 3a, 4a and Supplementary Figs. 3 and 6.

**Reporting summary**. Further information on research design is available in the Nature Research Reporting Summary linked to this article.

## Data availability

The 2010–2011 Namibian mobility data in the form of a geographic origin-destination adjacency matrix are freely available at: https://doi.org/10.1371/journal.pcbi.1004846.s002[34]. The 2013 Namibian Demographic and Health Survey (NDHS) data (Individual recode and HIV test results recode) are available for non-commercial use to registered users at: https://dhsprogram.com/data/dataset/Namibia_Standard-DHS_2013.cfm. The 2011 Namibia Population and Housing census data is freely available for non-commercial use to registered users at: https://nsa.org.na/microdata1/index.php/catalog/19.

## Code availability

Code used to carry out these analyses is available in a Github repository: https://github.com/eugenio-valdano/namibia_hiv_risk; https://doi.org/10.5281/zenodo.4651295[54].

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

## Acknowledgements

E.V., J.T.O., and S.B acknowledge the financial support of the National Institute of Allergy and Infectious Diseases, National Institutes of Health (grant R01 AI116493 and R56 AI152759). V.C. acknowledges the financial support of the Sorbonne Université Emergence project RISKFLOW.

## Author contributions

S.B., E.V., and V.C developed the concept. S.B. drafted the manuscript. E.V. performed the calculations. J.T.O. contributed to results. H.K.M. provided in-country expertise. All authors interpreted results and contributed to writing. All authors read and approved the final manuscript.

## Competing interests

The authors declare no competing interests.
