## [Peer Review File · Nature Communications]

Reviewers' comments:

Reviewer #1 (Remarks to the Author):

I have reviewed primarily the mobile phone data analysis in this study, which attempts to integrate mobility information to estimate the spatial spread of HIV in Namibia.

My biggest concern with this approach is that the mobility information used to infer transmission dynamics are from a single matrix published in 2016 by Ruktanonchai et al. This matrix quantifies the proportion of time that residents in each constituency spend in other constituencies. It has neither seasonality nor any demographic features, and in particular is not disaggregated by sex. This means that there is an underlying assumption that the measurements of residency among mobile phone users are the same for women and men.

Given the importance of the difference in prevalence between men and women, and the explicit goal here to understand the role of mobility in the overall distribution of HIV in men and women, the lack of disaggregated data is a significant problem. Men and women not only own mobile phones differentially, but are also expected to have very different patterns of travel for various reasons. This has been shown in a number of African settings (Camlin et al, 2019, Health and Place).

While aggregated CDR research products like the residency matrix used here are useful for general patterns of connectivity, I am concerned that for this purpose they are not sufficiently reliable or resolved. Presumably the results would be quite different depending on how trip lengths are distributed in the population, even within men. My expectation is that most travelers are men and not women, that this ratio increases with distance traveled, and that there is likely high variance in trip length in general (and probably quite important seasonal variation) that would significantly impact findings.

At the very least, a sensitivity analysis could be conducted to examine the impact of differential male:female travel measured using these data, and an exploration of how the distribution of trip lengths would impact expectations.

Overall, the study as it stands does not use CDRs appropriately for the stated scientific question and is not of sufficient quality to merit publication in Nature Communications without a major revision.

Reviewer #2 (Remarks to the Author):

This well-written manuscript by Valdano et al. presents an innovative analysis of population mobility in Namibia based on data from mobile phone call records, HIV prevalence and census information. The authors identify and model spatial networks of risk and propose to apply their findings on mobility-driven HIV acquisition to Namibia's strategic plans to eliminate HIV. The main limitations of this paper are the substantial temporal differences in data sources and the assumption that SIM cards represent individual residents of Namibia.

Specific comments

Authors:

- The author list does not appear to include any Namibian authors. Inclusion of an author from the Government of Namibia may provide valuable in-country perspective.

Introduction:

- Dates of datasets: In the introduction, the authors present a compelling case for the relevance of their work in the framework of the UNAIDS' 2030 treatment targets, as well as previous applications of mobile phone data for other infectious disease elimination initiatives. The CDR and HIV prevalence data described in the Introduction seem at first to have been collected from the same time period ("Using a mathematical framework, we integrate the CDR dataset with HIV-testing data, and census data, from Namibia"); only later in the paper (Discussion, page 12) does it become clear that these datasets were collected at substantively different points in time. The

authors should specify in the Introduction that the CDR data reflect mobile phone activity during 2010-2011, Namibia Demographic Health Survey data were collected in 2013, census data were collected in 2017 and treatment data were from 2018. This will allow the reader to interpret the context of the findings, especially regarding potential impact on HIV treatment policies and guidelines.

- SIM card users: The authors report CDR data from 1.6 million SIM card users in 2010-11. In 2010, however, the population of Namibia was 1.97 million, with 62% (~1.24 million) 15 years of age and older. Would be helpful for the authors to be explicit that there were more SIM cards than adults in 2010-2011 and they should therefore not suggest that each SIM card represents a resident of Namibia. This discrepancy suggests SIM card users may be from elsewhere, such as tourists and businesspeople from neighboring countries who obtained a local SIM card for their visit, etc. The authors might also add a sentence in the Discussion section about the characteristics of Namibian adults who did not own or use a cell phone in 2011, i.e., were missing from this analysis and what percentage of the population they accounted for in 2010. Residents of Namibia who did not use SIM cards may well have also contributed to the mobility and HIV networks described by the authors.

- Treatment equity: The authors describe the Government of the Republic of Namibia's current goal of reaching the UNAIDS treatment targets for 2030 and achieving treatment equity by geographic region. Treatment equity targets, however, go beyond geography and include age and gender targets, especially for young women. This is laid out explicitly in the COP19 report (ref #22). While the CDR data do not allow analysis of SIM card users, and thus mobile networks, by age and gender, the authors should acknowledge that treatment equity includes age and gender targets to contextualize the application of their results to geographic treatment targets.

- Other models of mobility: The authors discuss the network-based framework as the base of their models. Would be helpful if they added a sentence or two to explain how their framework adds to or diverges from other models of mobility, such as the work done in Rakai (ref#13) and others?

- Minor:

- o The authors refer to their data as "HIV-testing data", however, the term 'HIV testing data' refers to data on access to testing and testing behavior which does not seem to be the intended meaning. "HIV prevalence data" would be more appropriate.

- o On page 4 the authors mention the "most challenging regions of the country" but it's not clear what kind of challenges are being referred to. The text implies regions that are geographically inaccessible. It is important to note that the Government of Namibia is working to reach demographic groups such as young men who are often challenging to engage.

Results:

- To set the context of their results, the authors should define the term 'constituencies', as it is only first presented in the legend for Figure 6 in the Supplemental Material. The authors should explain that Namibia is divided into 14 geographic regions which are subdivided into 54 provinces which are then further subdivided into 121 constituencies.

- In the results, the authors found that Namibia is "highly mobile". How does the mobility compare with other countries?

- "The most vulnerable constituencies are the in-flow risk hubs..." How many of the 121 constituencies meet the risk hub criteria? A description of the characteristics of the risk hubs (urban/rural, size, proximity to national borders and flight patterns) would be helpful.

- Not clear why the authors selected Kongola and Rundu Urban as the examples of in-flow and out-flow risk hubs for men.

Discussion:

- In the discussion, the authors suggest that it is important to target the uninfected individuals within in-flow risk hubs. Since the findings about the specific risk hubs come from data that are ten years old, the authors may want to comment on the need to repeat these analyses with more recent data, setting aside privacy concerns,

Tables and Figures:

- Each legend should include the year in which the data are from, e.g. Figure 1b uses data from 2010-2011 and Figure 2b uses data from 2013.

- Figure 2b Legend refers to the unit of measure as "residents", but should be SIM card users

- Figure 3b refers to Kongola but it's not clear from other maps in the manuscript where Kongola is.

Response to Reviewer 1:

I have reviewed primarily the mobile phone data analysis in this study, which attempts to integrate mobility information to estimate the spatial spread of HIV in Namibia.

1. My biggest concern with this approach is that the mobility information used to infer transmission dynamics are from a single matrix published in 2016 by Ruktanonchai et al. This matrix quantifies the proportion of time that residents in each constituency spend in other constituencies. It has neither seasonality nor any demographic features, and in particular is not disaggregated by sex. This means that there is an underlying assumption that the measurements of residency among mobile phone users are the same for women and for men.

Population mobility networks are always modeled by aggregating large datasets of CDRs into Origin-Destination (OD) matrices. Algorithms are used to aggregate these data. The OD matrix that we used was constructed by Ruktanonchai and colleagues. This was constructed by aggregating CDRs from 9 billion calls/texts made from 1.19 million unique SIM cards over a twelve-month period (October 2010 to September 2011) in Namibia. The CDRs were aggregated using an algorithm that accounts for the time spent by each individual in each constituency. If we had accessed the individual-level CDRs, we would have had to aggregate them ourselves into an OD matrix. We make these points in our revised manuscript.

We would like to stress, that any mobility network is based on an OD matrix, and therefore the dataset of CDRs always needs to be aggregated. The appropriate level of aggregation of the dataset depends upon the question that is addressed. In our study, we are studying HIV risk over 12 months. It is appropriate to study at this time scale, as incidence rates are low, and human behavior is not seasonal (therefore transmission risk is not seasonal).

As we explain in more detail in our responses below, the lack of disaggregation of the data by gender, season, or trip length, is not a problem for the purpose of our study.

Notably, we provide below - in our response to point 2 and include in our revised manuscript - a data-based justification for the parsimonious assumption that we made that the measurements of residency are the same for women and for men.

2. Given the importance of the difference in prevalence between men and women, and the explicit goal here to understand the role of mobility in the overall distribution of HIV in men and women, the lack of disaggregated data is a significant problem. Men and women not only own mobile phones differentially,

but are also expected to have very different patterns of travel for various reasons. This has been shown in a number of African settings (Camlin et al. 2019, Health and Place).

We are aware of the excellent work by Dr. Camlin. I am currently her mentor on a faculty training grant and she is working on a similar analysis with me (as the one that we present in this analysis) on travel behavior and HIV.

We respectfully disagree with the Reviewer that the lack of disaggregated data, with respect to gender, is a significant problem. This disagreement is based on two assumptions that the Reviewer has made: (i) that men and women in Namibia own mobile phones differentially, and (ii) that men and women in Namibia have very different patterns of travel. Whilst these two assumptions are correct for some countries in Africa, they are not correct for others. In our revised manuscript, we present new data and analyses that show that these two assumptions are not correct for Namibia.

Regarding assumption (i): we have included data in our revised manuscript that show that there was not a significant gender difference in Namibia in the ownership of mobile phones (84.3% women, 84.2% men), the usage of mobile phones (93.0% (women), 92.8% (men)), or in the frequency of phone calls and texting (Supplementary Table 1).

Regarding assumption (ii): we have included data in our revised manuscript that show that women (15 to 49 years old) were slightly less likely to travel in terms of overnight trips than men: 37% of women versus 43% of men. However, similar proportions (18%) of each gender took long trips (trips that lasted for more than a month).

Taken together, our new analyses (included in our revised manuscript) provide support for using the aggregated data on CDRs (i.e., using the same Origin-Destination matrix for men and women in Namibia). We have also expanded the paragraph of the potential limitations of CDRs in the discussion.

3. While aggregated CDR research products like the residency matrix used here are useful for general patterns of connectivity, I am concerned that for this purpose they are not sufficiently reliable or resolved. Presumably the results would be quite different depending on how trip lengths are distributed in the population, even within men. My expectation is that most travelers are men and not women, that this ratio increases with distance traveled, and that there is likely high variance in trip length in general (and probably quite important seasonal variation) that would significantly impact findings.

The Reviewers concerns here, with the exception of trip length and seasonality, are addressed in point 2.

Regarding trip length: the Reviewer assumes that our results would be quite different depending on how trip lengths are distributed in the population. This is actually not the case. The variability in trip length has been accounted for by the algorithm that aggregated the data into the form of the OD matrix. Also, to the best of our knowledge, there is no known relationship between the length of a trip and the type of risk behavior; consequently, it is unnecessary to disaggregate the data based upon trip length.

The CDR data do show seasonal variation in mobility; this would be important to include if we were modeling malaria. However, this variation does not impact our findings. This is because HIV transmission occurs throughout the year; transmission is not seasonal.

We include our responses to point 3 in our revised manuscript.

4. At the very least, a sensitivity analysis could be conducted to examine the impact of differential male: female travel measured using these data, and an exploration of how the distribution of trip lengths would impact expectations.

A sensitivity analysis is not required, because, as explained in our responses to points 2 and 3: (i) there are no significant gender differences in travel, and (ii) variation in trip length does not impact our results.

Overall, the study as it stands does not use CDRs appropriately for the stated scientific question and is not of sufficient quality to merit publication in Nature Communications without a major revision.

In our revised manuscript, we have included additional data and analyses that demonstrate that our use of CDRs is appropriate for the scientific question that we address (please see responses to points 1-4). We hope that we have now addressed all of the concerns raised by this Reviewer regarding the disaggregation of the CDR dataset.

Response to Reviewer 2:

This well-written manuscript by Valdano et al. presents an innovative analysis of population mobility in Namibia based on data from mobile phone call records, HIV prevalence and census information. The authors identify and model spatial networks of risk and propose to apply their findings on mobility-driven HIV acquisition to Namibia's strategic plans to eliminate HIV.

We thank this Reviewer for their comments.

The main limitations of this paper are the substantial temporal differences in data sources and the assumption that SIM cards represent individual residents of Namibia.

We have now resolved the concern regarding the temporal differences in data sources (please see our response to point 2), and we have clarified our assumptions regarding SIM cards (please see our response to point 3).

Specific Comments:

1. Authors: The author list does not appear to include any Namibian authors. Inclusion of an author from the Government of Namibia may provide in-country perspective.

We very much appreciate this suggestion. We have added an author from Namibia, Dr. Kabwebwe Honoré Mitonga. Dr. Mitonga is the Associate Dean and the Head of the School of Public Health at the University of Namibia. His scientific expertise is in biostatistics and epidemiology. We have worked closely with Dr. Mitonga to revise our manuscript, he has made a substantial and very important contribution and provided an invaluable in-country perspective.

2. Dates of datasets: In the introduction, the authors present a compelling case for the relevance of their work in the framework of the UNAIDS' 2030 treatment targets, as well as previous applications of mobile phone data for other infectious disease elimination initiatives. The CDR and HIV prevalence data described in the Introduction seem at first to have been collected from the same time period ("Using a mathematical framework, we integrate the CDR dataset with HIV-testing data, and census data, from Namibia"); only later in the paper (Discussion, page 12) does it become clear that these datasets were collected at substantively different points in time. The authors should specify in the Introduction that the CDR data reflect mobile phone activity during 2010-2011, Namibia Demographic Health Survey (NDHS) data were collected in 2013, census data were collected in 2017 and treatment data were from 2018. This will allow the reader to interpret the context of the findings, especially regarding the potential impact on HIV treatment policies and guidelines.

In our revised manuscript, we do not use data that were collected at substantively different points in time. First, we wish to clarify that the census data that we use were collected in 2011 and not in 2017. Therefore, the census data were collected at the same time as the CDR data. We have removed the analysis of the treatment data that were collected in 2018. Therefore, all the data that we use in our revised manuscript were collected between 2010/11 and 2013. [The CDRs were collected in 2010/2011, the census data in 2011, and the prevalence data in 2013.] Notably, HIV prevalence in Namibia did not change significantly between 2013 and 2010/2011 when the CDR data were collected. In the introduction of our manuscript we now make this clear.

3. SIM card users: The authors report CDR data from 1.6 million SIM card users in 2010-11. In 2010, however, the population of Namibia was 1.97 million, with 62% (~1.24 million) 15 years of age and older. Would be helpful for the authors to be explicit that there were more SIM cards than adults in 2010-2011 and they should therefore not suggest that each SIM card represents a resident of Namibia. This discrepancy suggests SIM card users may be from elsewhere, such as tourists and businesspeople from neighboring countries who obtained a local SIM card for their visit, etc. The authors might also add a sentence in the Discussion section about the characteristics of Namibian adults who did not own or use a cell phone in 2011, i.e., were missing from this analysis and what percentage of the population they accounted for in 2010. Residents of Namibia who did not use SIM cards may well have also contributed to the mobility and HIV networks described by the authors.

We apologize, the Reviewer's concern is based on a typographical error in our manuscript. Our original manuscript stated that the dataset contained 1.55 SIM cards; this should have read 1.19 SIM cards. Using the correct number of SIM cards [and given that Namibia had a population of 1.34 million individuals (15 or older) in 2011 and 84% of the population owned mobile phones (and 93% used them)], it can be seen that relatively few of the 1.19 million SIM cards in the database would have been owned by non-Namibians. Therefore, movements of non-residents would have had little effect on the mobility network. As the Reviewer suggests, we now include this information in the revised manuscript and mention that some SIM card users would be tourists or business people.

As the Reviewer suggests we now include in the discussion section, a sentence describing the characteristics of Namibian adults who did not own or use a cell phone in 2011. We also state that 7% of Namibian adults were missing from our analysis.

As the Reviewer suggests we now mention in the discussion section that Namibians who did not use SIM cards may well have also contributed to the mobility and HIV networks.

4. Treatment equity: The authors describe the Government of the Republic of Namibia's current goal of reaching the UNAIDS treatment targets for 2030 and achieving treatment equity by geographic region. Treatment equity targets, however, go beyond geography and include age and gender targets, especially for young women. This is laid out explicitly in the COP19 report (ref #22). While the CDR data do not allow analysis of SIM card users, and thus mobile networks, by age and gender, the authors should acknowledge that treatment equity includes age and gender targets to contextualize the application of their results to geographic treatment targets.

As the Reviewer suggests we now explicitly state that treatment equity includes age and gender targets.

5. Other models of mobility: The authors discuss the network-based framework as the base of their models. Would be helpful if they added a sentence or two to explain how their framework adds to or diverges from other models of mobility, such as the work done in Rakai (ref#13) and others?

All previous work has focused on identifying small-scale mobility networks and analyzing their impact on localized HIV sub-epidemics; e.g., the authors of the Rakai study (ref#8 in the revision) identified a small-scale mobility network of individuals who moved amongst 38 communities. They analyzed the impact of this network on HIV transmission and dispersal within the localized sub-epidemic.

Our study is at a very different spatial scale: we identify very large-scale mobility networks that span an entire country and we analyze their impact on HIV transmission and dispersal at the national-level. By taking this large-scale approach we have gained a new conceptual understanding of the dynamics of generalized HIV epidemics: i.e., that HIV epidemics are large-scale complex networks of mobility-driven risk flows. Our novel results have significant implications for the control of HIV epidemics in sub-Saharan Africa. They imply that, to eliminate HIV, it is essential to design new control strategies that focus on disrupting these networks. As the Reviewer suggests, we have added this information to the revised manuscript.

Minor:

6. The authors refer to their data as "HIV-testing data", however, the term 'HIV testing data' refers to data on access to testing and testing behavior which does not seem to be the intended meaning. "HIV prevalence data" would be more appropriate.

We now refer to the data as HIV prevalence data.

7. On page 4 the authors mention the "most challenging regions of the country" but it's not clear what kind of challenges are being referred to. The text implies regions that are geographically inaccessible. It is

important to note that the Government of Namibia is working to reach demographic groups such as young men who are often challenging to engage.

We agree with the Reviewer; we have clarified that there are many different challenges that need to be met including geographic and demographic challenges.

8. To set the context of their results, the authors should define the term 'constituencies', as it is only first presented in the legend for Figure 6 in the Supplemental Material. The authors should explain that Namibia is divided into 14 geographic regions which are subdivided into 54 provinces which are then further subdivided into 121 constituencies.

We now do so.

9. In the results, the authors found that Namibia is "highly mobile". How does the mobility compare with other countries?

Unfortunately, we are not able to conduct a quantitative comparison as the necessary data do not exist.

10. "The most vulnerable constituencies are the in-flow risk hubs..." How many of the 121 constituencies meet the risk hub criteria? A description of the characteristics of the risk hubs (urban/rural, size, proximity to national borders and flight patterns) would be helpful.

We define a constituency as an in-flow risk hub if it is in the top 40% of constituencies with respect to the inflow of risks. In 2010/11, when the CDRs were collected, there were 107 constituencies, of which 96 had cell towers. 38 of the 96 constituencies are risk hubs. Each risk hub has a different amount of risk flowing into it; the greater the in-flow of risk, the more important the constituency is as a risk hub. We define the mechanisms that characterize in-flow risk hubs. We include this new information in our revised manuscript.

11. Not clear why the authors selected Kongola and Rundu Urban as the examples of in-flow and out-flow risk hubs for men.

We chose these two constituencies as examples simply due to the clarity of their visualization as networks.

12. In the discussion, the authors suggest that it is important to target the uninfected individuals within in-flow risk hubs. Since the findings about the specific risk hubs come from data that are ten years old, the

authors may want to comment on the need to repeat these analyses with more recent data, setting aside privacy concerns.

We now do this.

13. Tables and Figures: Each legend should include the year in which the data are from, e.g. Figure 1b uses data from 2010-2011 and Figure 2b uses data from 2013.

We now do this.

14. Figure 2b Legend refers to the unit of measure as “residents”, but should be SIM card users.

We believe the Reviewer is referring to Figure 2a. We choose to leave the unit of measure as “residents” as this is the terminology used in the Origin-Destination matrix that we received.

15. Figure 3b refers to Kongola but it’s not clear from other maps in the manuscript where Kongola is.

We have now marked Kongola on the map that is shown in Supplementary Fig. 5.

REVIEWERS' COMMENTS

Reviewer #2 (Remarks to the Author):

MANUSCRIPT #: NCOMMS-20-27610

Valdano et al. have revised their manuscript in response to comments. They have addressed the main limitations noted in the prior review, i.e., temporal differences in data sources and the assumption that SIM cards represent individual residents of Namibia. They have added an author, Dr. Mitonga, from the Government of Namibia who has relevant technical expertise and the authors note that Dr. Mitonga has added his perspective.

The abstract does not mention that the analyses are based on data from 2010-2011. This omission needs to be corrected.

The authors have added some text about how their work differs from the findings on mobility networks in Rakai. The authors feel their work is different because it is at a larger spatial scale but scale alone does not describe all differences. There may be differences in theoretical models as well as the key findings and conclusions. The paper would be stronger if the authors explained how their work builds on or complements or refutes the Rakai work.

While the authors work addresses generalized epidemics which are largely driven by heterosexual sex, it would be reasonable to acknowledge that their work does not account for networks of men who have sex with men and these networks may add to the HIV epidemic in Namibia.

The role of Angola in the mobility network is not mentioned even though most (7 of 10) outflow hubs and all inflow hubs are on or very near the Namibia-Angola border.

Finally, although that Government of Namibia did not account for the 2010-2011 mobility networks in its allocation of antiretrovirals and other HIV response resources, by 2017, as noted by the authors, Namibia had done very well in responding to its epidemic, with "...~83% of all HIV-infected adults in Namibia [on] treatment." The authors do not point out that this level of treatment coverage in 2017 was very high, and higher than what most other sub-Saharan African countries with severe generalized epidemics were able to achieve. Clearly, in 2017, Namibia's HIV response program was doing well even without using mobility network analyses.

The authors state that to continue to make progress toward the 2030 targets, analyses of mobility networks will be needed ("As HIV approaches elimination, mobility-driven transmission is likely to become increasingly important.") The logic is clear but the evidence for this argument comes from polio and malaria response programs. In view of the lack of strong evidence, the conclusion of the Abstract ("Large-scale networks of mobility-driven risk flows underlie generalized HIV epidemics in sub-Saharan Africa. Elimination strategies need to target mobility-driven transmission.") should be moderated.

Minor:

Tables and Figures: The authors state that they have now included information in each figure legend about the year the data are from; this has been done, however, only for Figures 2a-c.

Reviewer #3 (Remarks to the Author):

Title: Using mobile phone data to identify spatial risk flow networks within an HIV epidemic in sub-Saharan Africa: implications for elimination

Authors: Eugenio Valdano¹, Justin T. Okano¹, Vittoria Colizza², Honore K. Mitonga³, Sally Blower^{1*}

Recommendation: Minor revision

Comments:

I was asked to provide an evaluation of the authors' responses to the original Reviewer #1's comments, as Reviewer #1 was not available to re-review the manuscript; I have limited my

review to these previously identified content areas. Below, I have provided additional comments for the four main concerns of reviewer 1 (and the authors' replies), numbered in the same order as before.

Comment #1: regarding use of a single origin-destination matrix not disaggregated for seasonality nor gender.

I appreciate the additions of sensitivity analyses and more detailed descriptions of the limitations of the lack of gender-specific OD matrices that the authors have provided. These sensitivity analyses suggest that phone ownership, usage, and length of trips (at the national level) did not substantially vary between men and women, though men may have taken slightly longer trips. Of course, even if men and women travel with the same frequency and same duration of trips, their origins and destinations could still be different (i.e. their "true" OD matrices may be different). The authors appropriately recognize this underlying limitation in their discussion.

The authors state that differential travel patterns by gender would not substantively change their main conclusions, which may be true. The results of the paper, however, do present a broad variety of results and conclusions which are gender-specific, however. I therefore agree in general with the original Reviewer #1 that it would be ideal to have gender-specific OD matrices in order to reflect any differences in travel patterns when estimating risk flows and the other gender-specific results of this paper, and that this remains an important limitation of the manuscript. As the authors point out, however, the lack of gender disaggregation is in fact a limitation of the underlying CDRs used to construct the OD matrices, so there is only so much that the authors could do to try to address this limitation. Their approach – clearly explaining the limitation and analyzing auxiliary data sources to help provide context for this assumption – broadly seems reasonable.

My one additional question is whether the authors would be able to disaggregate the analysis of travel data from the 2013 NDHS by region of residence. I did not see this particular analysis in the 2013 NDHS full report, though may have missed it. If there are certain regions where duration of trip or number of trips taken varies substantially by gender, this could support Reviewer #1's concern about the lack of gender disaggregation in the OD matrices and would warrant further discussion as a limitation in the article. For instance, if men in region A travel more frequently than women, and women in region B travel more frequently than men, you might still see that these subnational differences generally cancel each other out when aggregating to the national level. A pattern like this, however, would suggest that there are in fact differences in the "true" OD matrices for men and women

If this disaggregation by region is not possible due to limitations of the underlying NDHS data, however, then my opinion is that the sensitivity analyses and revised discussion are appropriate, as long as this limitation is highlighted in the discussion.

In terms of seasonality: I generally agree with the authors' rebuttal that – while seasonal patterns may exist in the CDR data – seasonality is unlikely to be an important contributor in this analysis. In essence, the authors are comparing prevalence data collected in May-Sept 2013 in the 2013 NDHS to mobile phone data collected between October 2010-Sept 2011. The implicit assumptions here are two-fold, as I see them: 1) prevalence data collected in May-Sept is generally representative of prevalence for a full year, and 2) there were not substantial changes in mobility patterns or HIV prevalence between 2010-11 and 2013.

The authors state that human sexual behavior is not seasonal, which may be true (I am not an expert in this area; if there is a citation for this statement would be useful to include). More importantly, however, the authors are examining the flow of populations throughout the year with respect to the *prevalence* of HIV in origin and destination locations. One could hypothesize a potential seasonal variation in HIV *incidence* for a given location – for instance, driven by seasonality in mobility patterns -- though this may or may not actually be true. HIV *prevalence*, however, is more likely to reflect longer-term HIV transmission trends and less likely to be substantially seasonal in nature, especially where incidence is relatively low in comparison to prevalence. It therefore seems reasonable to assume that HIV prevalence data collected in May-Sept 2013 should be representative of the full-year 2013 HIV prevalence for each location. The authors' approach and responses with respect to seasonality, therefore, seem reasonable to

me. My one suggestion would be for the authors to rely less on the statement that human sexual behavior is not seasonal (this seems like it would be more likely to explain a lack of seasonality in incidence), and more on the observation that in a low-incidence setting, HIV prevalence is unlikely to be substantially seasonal. The latter seems to be a more salient justification for the authors' approach. Both of these points are made in the authors' rebuttal, but only the point re: seasonality of human sexual behavior is included on page 10 of the revised manuscript.

As for the second implicit assumption (i.e. that the comparison between mobility data collected in 2010-11 and prevalence data in 2013 is valid); I would defer to Reviewer #2 who had comments relating to this.

Comment #2: additional comments regarding challenges presented by a lack of gender-disaggregated CDR data / OD matrices.

For the first concern – whether there are gender differences in mobile phone usage or ownership in Namibia – the authors' analysis showing similar patterns of mobile phone ownership and usage from the Afrobarometer surveys is extremely helpful and compelling. In my view, this appropriately addresses Reviewer #1's concerns re: potential variation in mobile phone ownership or usage by gender.

For the second concern, as noted above, I think that the authors have generally addressed this appropriately in their revised manuscript (see above for question re: whether these analyses of NDHS data and travel patterns could be disaggregated by region). The supplementary analyses regarding general travel patterns are supportive of their approach, and they appropriately recognize in the discussion that there is a persistent limitation in this area (i.e. that even if the number and length of trips are similar between men and women, the origins and destinations may still differ).

Comment #3: concerns regarding the reliability and resolution of the OD matrices used, and comment #4, regarding a sensitivity analysis.

See above for my comments regarding the authors' responses to Reviewer #1's concerns about seasonality and differences by gender. I also appreciate the authors' comments that the algorithm used by Ruktanonchai et al accounts for the time spent by each individual in a given consistency (also on page 4 of the manuscript). This, in combination with the 2013 NDHS analysis showing a that trip length was similar between males and females, appropriately addresses the concerns regarding trip length.

In the absence of additional gender-disaggregated CDR data, I'm not sure how the authors could properly parameterize a sensitivity analysis as suggested by Reviewer #1 in their comment #4, given a lack of underlying gender-disaggregated data. If there are substantial regional differences in the NDHS data for trip frequency or duration, then perhaps this could be used to parameterize a sensitivity analysis (i.e. by using the regional gender patterns in the NDHS data to derive hypothetical gender-specific OD matrices from the both-gender OD matrix). Unless there are substantial regional differences in travel patterns by gender in the NDHS data, however, I do not feel that this is obligatory.

REVIEWERS' COMMENTS

Reviewer #2 (Remarks to the Author):

MANUSCRIPT #: NCOMMS-20-27610

Valdano et al. have revised their manuscript in response to comments. They have addressed the main limitations noted in the prior review, i.e., temporal differences in data sources and the assumption that SIM cards represent individual residents of Namibia. They have added an author, Dr. Mitonga, from the Government of Namibia who has relevant technical expertise and the authors note that Dr. Mitonga has added his perspective.

The abstract does not mention that the analyses are based on data from 2010-2011. This omission needs to be corrected.

We respectfully disagree with the reviewer, because we believe that it may be confusing to readers to list these dates in the abstract without explaining why we are using data from 2010-2011. We would need to explain that we are using data from 2010-2011 because we are reconstructing the epidemic of a decade ago. We would also need to explain that we are reconstructing this epidemic so that we can determine the effect of mobility on the epidemic without including the confounding effect of treatment on the epidemic. While all this is explained in detail in the manuscript, it would exceed the word limit for the abstract.

The authors have added some text about how their work differs from the findings on mobility networks in Rakai. The authors feel their work is different because it is at a larger spatial scale but scale alone does not describe all differences. There may be differences in theoretical models as well as the key findings and conclusions. The paper would be stronger if the authors explained how their work builds on or complements or refutes the Rakai work.

The studies that have been conducted in Rakai are excellent; we cite two recent studies in our manuscript. Our work differs a great deal from the Rakai study, besides in spatial scale. We focus on mobility in terms of "circular travel" (people travel to a destination and then return to where they live), the Rakai studies focus on mobility in terms of permanent migration (people move permanently from one location to another). We present theoretical models, the Rakai studies present statistical analyses of phylogenetic and epidemiologic data. Our work focuses on spatial networks, the Rakai studies focus on sexual networks. As such, our work and the Rakai studies are very different. The Rakai study, our study – and many other studies – have contributed to the current understanding of the impact of mobility and migration on HIV epidemics. We now acknowledge the important contributions of the Rakai studies (and several other studies) to this body of knowledge by including the following sentence in the first paragraph of the discussion:

Previous studies have made important contributions to our understanding of the impact of mobility on transmission at the community level⁶⁻¹².

While the authors work addresses generalized epidemics which are largely driven by sexual heterosexual sex, it would be reasonable to acknowledge that their work does not account for networks of men who have sex with men and these networks may add to the HIV epidemic in Namibia.

We agree with the Reviewer, and have added the following sentences to the paragraph in the Discussion that describes the limitations of our study:

Finally, our study focuses on identifying spatial networks of risk flows in a generalized epidemic where the vast majority of transmission is through heterosexual sex. We have not modeled spatial networks of risk flows among men who have sex with men. These networks could be included in future studies.

The role of Angola in the mobility network is not mentioned even though most (7 of 10) outflow hubs and

all inflow hubs are on or very near the Namibia-Angola border.

The Reviewer raises an important point. We have added the following sentences to the Discussion:

Many in-flow and out-flow hubs in Namibia are close to the Angolan border. HIV prevalence along the border (in adults aged 15-49 years old) is substantially lower in Angola (5-6%) than in Namibia (9-32%). This suggests that the out-flow of risk from Namibia would have been greater than the in-flow of risk from Angola. Consequently, Namibia would have had more of an impact on the HIV epidemic in Angola, than Angola would have had on the HIV epidemic in Namibia.

Finally, although that Government of Namibia did not account for the 2010-2011 mobility networks in its allocation of antiretrovirals and other HIV response resources, by 2017, as noted by the authors, Namibia had done very well in responding to its epidemic, with "...~83% of all HIV-infected adults in Namibia [on] treatment." The authors do not point out that this level of treatment coverage in 2017 was very high, and higher than what most other sub Saharan African countries with severe generalized epidemics were able to achieve. Clearly, in 2017, Namibia's HIV response program was doing well even without using mobility network analyses.

We agree with this important point made by the Reviewer. We have added the following sentence to our manuscript:

By 2017, the HIV treatment program in Namibia was doing extremely well: ~83% of all HIV-infected adults in Namibia were on treatment, a level of coverage that is among the highest in Africa.

The authors state that to continue to make progress toward the 2030 targets, analyses of mobility networks will be needed ("As HIV approaches elimination, mobility-driven transmission is likely to become increasingly important.") The logic is clear but the evidence for this argument comes from polio and malaria response programs. In view of the lack of strong evidence, the conclusion of the Abstract ("Large-scale networks of mobility-driven risk flows underlie generalized HIV epidemics in sub-Saharan Africa. Elimination strategies need to target mobility-driven transmission.") should be moderated.

We have done as the Reviewer has requested, the last sentence of the abstract now reads:

In order to eliminate HIV, it is likely to become increasingly important to implement innovative control strategies that focus on disrupting risk flows.

Minor:

Tables and Figures: The authors state that they have now included information in each figure legend about the year the data are from; this has been done, however, only for Figures 2a-c.

We have included the dates that the data were collected in Figures 2a-c as the empirical data are shown in these Figures. However, the empirical data are not shown in the other Figures - these Figures show the results of our modeling - therefore we do not include the dates of data collection.

Reviewer #3 (Remarks to the Author):

Title: Using mobile phone data to identify spatial risk flow networks within an HIV epidemic in sub-Saharan Africa: implications for elimination

Authors: Eugenio Valdano¹, Justin T. Okano¹, Vittoria Colizza², Honore K. Mitonga³, Sally Blower^{1*}

Recommendation: Minor revision

Comments:

I was asked to provide an evaluation of the authors' responses to the original Reviewer #1's comments, as Reviewer #1 was not available to re-review the manuscript; I have limited my review to these previously identified content areas. Below, I have provided additional comments for the four main concerns of reviewer 1 (and the authors' replies), numbered in the same order as before.

Comment #1: regarding use of a single origin-destination matrix not disaggregated for seasonality nor gender. I appreciate the additions of sensitivity analyses and more detailed descriptions of the limitations of the lack of gender-specific OD matrices that the authors have provided. These sensitivity analyses suggest that phone ownership, usage, and length of trips (at the national level) did not substantially vary between men and women, though men may have taken slightly longer trips. Of course, even if men and women travel with the same frequency and same duration of trips, their origins and destinations could still be different (i.e. their "true" OD matrices may be different). The authors appropriately recognize this underlying limitation in their discussion.

Thank you, we are happy that the Reviewer is satisfied with how we have addressed this concern.

The authors state that differential travel patterns by gender would not substantively change their main conclusions, which may be true. The results of the paper, however, do present a broad variety of results and conclusions which are gender-specific, however. I therefore agree in general with the original Reviewer #1 that it would be ideal to have gender-specific OD matrices in order to reflect any differences in travel patterns when estimating risk flows and the other gender-specific results of this paper, and that this remains an important limitation of the manuscript. As the authors point out, however, the lack of gender disaggregation is in fact a limitation of the underlying CDRs used to construct the OD matrices, so there is only so much that the authors could do to try to address this limitation. Their approach – clearly explaining the limitation and analyzing auxiliary data sources to help provide context for this assumption – broadly seems reasonable.

Thank you, we are happy that the Reviewer is satisfied with how we have addressed this concern.

My one additional question is whether the authors would be able to disaggregate the analysis of travel data from the 2013 NDHS by region of residence. I did not see this particular analysis in the 2013 NDHS full report, though may have missed it. If there are certain regions where duration of trip or number of trips taken varies substantially by gender, this could support Reviewer #1's concern about the lack of gender disaggregation in the OD matrices and would warrant further discussion as a limitation in the article. For instance, if men in region A travel more frequently than women, and women in region B travel more frequently than men, you might still see that these subnational differences generally cancel each other out when aggregating to the national level. A pattern like this, however, would suggest that there are in fact differences in the "true" OD matrices for men and women. If this disaggregation by region is not possible due to limitations of the underlying NDHS data, however, then my opinion is that the sensitivity analyses and revised discussion are appropriate, as long as this limitation is highlighted in the discussion.

We thank the Reviewer for their thoughtful comment. Unfortunately, it is not possible to disaggregate the travel data from the 2013 NDHS by constituency which would be necessary to construct gender-specific OD matrices. We therefore agree with the Reviewer that this is a limitation of our study which should be mentioned in the discussion. We have now done this by including the following sentences in the Discussion:

The data that we have presented have shown that phone ownership and usage and length of trips did not differ by gender. However, men traveled slightly more frequently than women, and we do not know whether there were gender differences in the origins and destinations of trips. Therefore, it is possible that women and men had different mobility networks. If true, this could potentially change which of the constituencies were risk hubs.

In terms of seasonality: I generally agree with the authors' rebuttal that – while seasonal patterns may exist in the CDR data – seasonality is unlikely to be an important contributor in this analysis. In essence, the authors are comparing prevalence data collected in May-Sept 2013 in the 2013 NDHS to mobile phone data collected between October 2010-Sept 2011. The implicit assumptions here are two-fold, as I see them: 1) prevalence data collected in May-Sept is generally representative of prevalence for a full year, and 2) there were not substantial changes in mobility patterns or HIV prevalence between 2010-11 and 2013.

The authors state that human sexual behavior is not seasonal, which may be true (I am not an expert in this area; if there is a citation for this statement would be useful to include). More importantly, however, the authors are examining the flow of populations throughout the year with respect to the *prevalence* of HIV in origin and destination locations. One could hypothesize a potential seasonal variation in HIV *incidence* for a given location - for instance, driven by seasonality in mobility patterns -- though this may or may not actually be true. HIV *prevalence*, however, is more likely to reflect longer-term HIV transmission trends and less likely to be substantially seasonal in nature, especially where incidence is relatively low in comparison to prevalence. It therefore seems reasonable to assume that HIV prevalence data collected in May-Sept 2013 should be representative of the full-year 2013 HIV prevalence for each location. The authors' approach and responses with respect to seasonality, therefore, seem reasonable to me. My one suggestion would be for the authors to rely less on the statement that human sexual behavior is not seasonal (this seems like it would be more likely to explain a lack of seasonality in incidence), and more on the observation that in a low-incidence setting, HIV prevalence is unlikely to be substantially seasonal. The latter seems to be a more salient justification for the authors' approach. Both of these points are made in the authors' rebuttal, but only the point re: seasonality of human sexual behavior is included on page 10 of the revised manuscript.

We thank the reviewer for their constructive comments. We looked for references on the lack of seasonality of human sexual behavior but were unable to find any. We agree with the Reviewer that even in the case of seasonal changes in sexual behavior which could induce variations in incidence, these variations would have a negligible impact on prevalence each year, given that incidence is much lower than prevalence. We have edited the relevant paragraph as suggested by the Reviewer. This now reads:

It is not necessary to model seasonal changes in mobility as HIV transmission occurs throughout the year. Even a substantial seasonal variation in transmission would have a negligible impact (over a year) on prevalence; this is because prevalence is an order of magnitude higher than incidence.

As for the second implicit assumption (i.e. that the comparison between mobility data collected in 2010-11 and prevalence data in 2013 is valid); I would defer to Reviewer #2 who had comments relating to this.

Thank you. We note that Reviewer #2 says that they are satisfied with how we have addressed this concern.

Comment #2: additional comments regarding challenges presented by a lack of gender-disaggregated CDR data / OD matrices. For the first concern – whether there are gender differences in mobile phone usage or ownership in Namibia – the authors' analysis showing similar patterns of mobile phone ownership and usage from the Afrobarometer surveys is extremely helpful and compelling. In my view, this appropriately addresses Reviewer #1's concerns re: potential variation in mobile phone ownership or usage by gender.

Thank you, we are happy that the Reviewer is satisfied with how we have addressed this concern.

For the second concern, as noted above, I think that the authors have generally addressed this appropriately in their revised manuscript (see above for question re: whether these analyses of NDHS data and travel patterns could be disaggregated by region). The supplementary analyses regarding general travel patterns are supportive of their approach, and they appropriately recognize in the discussion that there is a persistent limitation in this area (i.e. that even if the number and length of trips are similar between men and women, the origins and destinations may still differ).

Thank you, we are happy that the Reviewer is satisfied with how we have addressed this concern.

Comment #3: concerns regarding the reliability and resolution of the OD matrices used, and comment #4, regarding a sensitivity analysis.

See above for my comments regarding the authors' responses to Reviewer #1's concerns about seasonality and differences by gender. I also appreciate the authors' comments that the algorithm used by Ruktanonchai et al accounts for the time spent by each individual in a given consistency (also on page 4 of the manuscript). This, in combination with the 2013 NDHS analysis showing a that trip length was similar between males and females, appropriately addresses the concerns regarding trip length.

Thank you, we are happy that the Reviewer is satisfied with how we have addressed this concern.

In the absence of additional gender-disaggregated CDR data, I'm not sure how the authors could properly parameterize a sensitivity analysis as suggested by Reviewer #1 in their comment #4, given a lack of underlying gender-disaggregated data. If there are substantial regional differences in the NDHS data for trip frequency or duration, then perhaps this could be used to parameterize a sensitivity analysis (i.e. by using the regional gender patterns in the NDHS data to derive hypothetical gender-specific OD matrices from the both-gender OD matrix). Unless there are substantial regional differences in travel patterns by gender in the NDHS data, however, I do not feel that this is obligatory.

We thank the Reviewer for this positive assessment. In addition, as described in the response to a previous point, we have edited the relevant section in the discussion, to better discuss the limitations of our lack of gender-stratified mobility data.